# Biochemical Parameters of Female Wistar Rats and Their Offspring Exposed to Inorganic Mercury in Drinking Water during the Gestational and Lactational Periods

**DOI:** 10.3390/toxics10110664

**Published:** 2022-11-05

**Authors:** Maria Eduarda A. Galiciolli, Taíse F. Pedroso, Mariana Mesquita, Vitor A. Oliveira, Maria E. Pereira, Cláudia S. Oliveira

**Affiliations:** 1Instituto de Pesquisa Pelé Pequeno Príncipe, Rua Silva Jardim, 1632, Curitiba 80250-060, PR, Brazil; 2Faculdades Pequeno Príncipe, Avenida Iguaçu, 333, Curitiba 80230-020, PR, Brazil; 3Programa de Pós-Graduação em Ciências Biológicas: Bioquímica Toxicológica, Universidade Federal de Santa Maria, Santa Maria 97105-900, RS, Brazil

**Keywords:** breast milk, inorganic mercury, in utero, metallothionein, offspring, PBG-synthase

## Abstract

The aim of this study was to investigate the effects of inorganic mercury (Hg^2+^) exposure on biochemical parameters of dams and their offspring exposed to metal in drinking water. Female Wistar rats were exposed to 0, 10, and 50 µg Hg^2+^/mL (as HgCl_2_) for 42 days corresponding to gestational (21 days) and lactational (21 days) periods. The offspring were sacrificed on postnatal days 10, 20, 30, and 40. Dams exposed to Hg^2+^ presented a decrease in water intake in gestation [total: F(2,19) = 15.84; *p* ≤ 0.0001; daily: F(2,21) = 12.71; *p* = 0.0002] and lactation [total: F(2,19) = 4.619; *p* = 0.024; daily: F(2,21) = 5.309; *p* = 0.0136] without alteration in food intake. Dams exposed to 50 µg Hg^2+^/mL had an increase in kidney total [F(2,21) = 8.081; *p* = 0.0025] and relative [F(2,21) = 14.11; *p* = 0.0001] weight without changes in biochemical markers of nephrotoxicity. Moreover, dams had an increase in hepatic [F(2,10) = 3.847; *p* = 0.0577] and renal [F(2,11) = 6.267; *p* = 0.0152] metallothionein content concomitantly with an increase in renal Hg levels after Hg^2+^ exposure. Regarding offspring, the exposure to Hg^2+^  *in utero* and breast milk increased the relative liver [F(2,18) = 5.33; *p* = 0.0152] and kidney [F(2,18) = 3.819; *p* = 0.0415] weight only on the postnatal day 40. In conclusion, dams were able to handle the Hg^2+^ avoiding the classic Hg^2+^ toxic effects as well as protecting the offspring. We suggest that this protection is related to the hepatic and renal metallothionein content increase.

## 1. Introduction

Mercury (Hg), a non-essential metal, is naturally (via volcanism and earth crust erosion) and anthropogenically (as a byproduct of fuel combustion, cement production, metals smelting, and silver and gold extraction) released into the environment. Apart from this, the uncontrolled use and discard of Hg-containing products (lamps, batteries, paints, fungicides, and medical instruments) have been enhancing the environmental Hg levels [1,2]. Humans can be exposed to Hg occupationally [3,4,5] or through the intake of contaminated food or water [5,6,7]. To avoid the consequences of the exposure, the Environmental Protection Agency (EPA) established that the concentration of mercury in drinking water must not exceed 0.002 µg/mL [8].

Once exposed to Hg, humans and non-human animals have symptoms directly related to the Hg chemical form. The central nervous system is more susceptible to organic Hg exposure [2,6,9,10] and the renal system is more susceptible to inorganic Hg exposure [11,12,13,14]. However, the Hg chemical forms (organic and inorganic) also cause toxic effects on the hematologic, hepatic, cardiovascular, and reproductive systems [15].

Although most of the cases of Hg contamination occur after the exposure to the organic (diet) and elemental (occupational) Hg forms, studies with inorganic Hg are necessary, since the organic and elemental Hg can be interconverted by methylation/demethylation or oxidation/reduction processes in the environment [16,17,18] and the body [19]. For instance, elemental mercury (Hg^0^) is converted to Hg^2+^ by the catalase enzyme [20], and methylmercury (MeHg^+^) is demethylated, mainly in the gastrointestinal tract, releasing ions Hg^2+^ [6,19]. Thus, the Hg chemical form that the organism is exposed to is not necessarily the same chemical form that reaches the tissues and causes the damage. Several studies have demonstrated inorganic Hg accumulation in rat tissues after MeHg^+^ exposure [21,22,23,24].

As observed in Minamata disease and other cases of Hg contamination, fetuses and infants are more sensitive to Hg exposure than adults [25,26,27,28]. The high sensitivity of young individuals compared to adults may be related to the immaturity of organs and membranes and their inability to properly process toxic agents [29]. Several studies have reported the toxicity caused by organic Hg exposure during pre- and/or postnatal periods [30,31,32,33,34,35,36,37,38]. However, few studies are about inorganic Hg exposure during pré- and/or postnatal periods [39,40,41,42,43].

Since the organic and elemental Hg forms can be converted to inorganic Hg, and animals in development are vulnerable to exogenous substances exposure, more studies using the inorganic Hg, as a toxic agent, during the developmental period are necessary. Thus, this work aimed to evaluate biochemical markers of toxicity in dams exposed to different doses of inorganic Hg in drinking water during the gestational and lactational periods, as well as to evaluate the same biochemical markers in the offspring. Additionally, the role of the metallothioneins in dams’ Hg sensitivity was investigated. 

## 2. Materials and Methods

### 2.1. Chemicals

Mercuric chloride, sodium chloride, potassium phosphate monobasic and dibasic, absolute ethanol, sodium hydroxide, trichloroacetic acid, ο-phosphoric acid, perchloric acid, glacial acetic acid, ß-mercaptoethanol, sucrose, hydrochloric acid, phenylmethylsulphonylfluoride (PMSF), chloroform, calcium disodium ethylenediaminetetraacetate (EDTA), 5,5-dithio-*bis*(2-nitrobenzoic acid) (DTNB), and tris(hydroxymethyl) aminomethane were purchased from Merck (Darmstadt, Germany). Bovine serum albumin and Coomassie brilliant blue g were obtained from Sigma (St. Louis, MO, USA). ρ-Dimethylaminobenzaldehyde was obtained from Riedel (Seelze, Han, Germany). The commercial kits for biochemical dosages were obtained from Kovalent do Brasil Ltd.a. (São Gonçalo, RJ, Brazil) or Labtest Diagnóstica S.A. (Lagoa Santa, MG, Brazil).

### 2.2. Animals

Forty Wistar rats (30 virgin female rats, 220 ± 20 g; and 10 mature male rats, 300 ± 50 g) obtained from the Animal Facility of the Federal University of Santa Maria, Brazil, were transferred to our breeding colony and maintained on a 12 h light/dark cycle and controlled temperature (22 ± 2 °C). The animals had free access to water and commercial food. Studies were conducted following the national and institutional guidelines (University Ethics Committee Guidelines, Process number 096/2011) for experiments with animals.

### 2.3. Mating

After 2 weeks of adaptation, the female and male rats (3:1) were placed in the same cage for mating as described by Oliveira C.S. et al. [41,43]. Mating was confirmed by the presence of sperm in vaginal smears. The day when the sperm was detected was considered as day 0 of gestation (GD 0).

### 2.4. Treatment

Pregnant rats were randomly divided into three groups. The experiment started with 30 female rats (*n* = 10 per group); however, three rats from the control group (0), one rat from group 10 (10), and three rats from group 50 (50) were not pregnant or delivered less than eight pups. Thus, the final experimental number used in this work is described below: 

0 (*n* = 7): drinking water;

10 (*n* = 9): 10 µg Hg^2+^/mL in drinking water;

50 (*n* = 7): 50 µg Hg^2+^/mL in drinking water.

Inorganic mercury doses (10 and 50 µg Hg^2+^/mL) were chosen based on a previous study from our research team [41], in which the concentrations of 0.2, 0.5, 10, and 50 µg Hg^2+^/mL were tested for 20 days and only the two highest doses caused significant alteration in animals body weight gain and food intake [41]. HgCl_2_ was dissolved in distilled water and provided to dams in drinking water. 

The dams were placed individually in polycarbonate cages and exposed to different doses of HgCl_2_ from gestation day 0 until lactation day 21 (~42 days of exposure). Every 2 days, the dams were weighed, and the food and water intake were quantified and replaced. On postnatal day 1, the litters were standardized in eight pups to avoid undernutrition effects. Dams were sacrificed on day 21 of lactation, and every 10 postnatal days (postnatal days: 10, 20, 30, and 40) two newborns per litter were sacrificed. Blood, liver, kidney, and brain were collected to biochemical assays and metal quantification.

### 2.5. Biochemical Assays

#### 2.5.1. Urea and Creatinine

For urea and creatinine analyses, serum was obtained by total blood centrifugation at 3000 g for 10 min and frozen until the analyses. Urea and creatinine levels were determined using commercial kits (Labtest). 

#### 2.5.2. Porphobilinogen Synthase (PBG-Synthase) Activity

For PBG-synthase activity assay, the liver, kidneys, and brain were quickly removed, placed on ice, and homogenized in 7, 5, and 3 volumes, respectively, of NaCl (150 mmol/L, pH 7.4). The homogenate was centrifuged at 8000 g for 30 min at 4 °C and the supernatant fraction was used in the enzymatic assay. The PBG-synthase activity was measured according to the method of Sassa [44], as modified by Peixoto et al. [45]. The enzymatic activity was expressed as nmol of porphobilinogen (PBG) formed per hour per mg protein. Protein concentrations were determined by the Coomassie blue method [46] using bovine serum albumin as standard.

#### 2.5.3. Thiol Groups

For total and non-protein thiol (-SH) analyses, tissues were quickly removed, placed on ice, and homogenized in 5 volumes of Tris–HCl buffer (10 mM, pH 7.4). The homogenate was centrifuged at 1050 g for 20 min at 4 °C. The supernatant fraction (S1) was used for the analyses. To determinate total thiol levels, the S1 was used, and to determine non-protein thiol (NPSH) levels the protein fraction of S1 was precipitated with 200 µL of 4% trichloroacetic acid followed by centrifugation (1050× *g*, 10 min). Thiol groups were determined by Ellman’s reagent [47]. A standard curve using reduced glutathione (GSH) as reference was constructed to express the total and non-protein thiol levels in µg of SH per gram of wet tissue.

#### 2.5.4. Metallothionein (MT) Levels

For metallothionein levels assay, the liver and kidneys were homogenized in 4 volumes of 20 mM Tris-HCl buffer, pH 8.6, containing 0.5 mM PMSF as antiproteolytic agent and 0.01% β-mercaptoethanol as a reducing agent. The homogenate was centrifuged at 16,000× *g* for 30 min. Aliquots of 1 mL of supernatant were mixed with 1.05 mL of cold (−20 °C) absolute ethanol and 80 µL of chloroform; the samples were then centrifuged at 6000× *g* for 10 min. The collected supernatant was combined with 3 volumes of cold ethanol (−20 °C), incubated at −20 °C for 1 h, and centrifuged at 6000× *g* for 10 min. The metallothionein-containing pellets were then rinsed with a solution containing 87% ethanol and 1% chloroform and centrifuged at 6000× *g* for 10 min. The pellet was resuspended in 150 µL 0.25 M NaCl and subsequently 150 µL 1 N HCl containing 4 mM EDTA was added to the sample.

Metallothionein levels were determined by the colorimetric method using Ellman’s reagent [47] as described by Viarengo et al. [48] and Peixoto et al. [45]. A standard curve using GSH as reference was constructed to express the MT levels in µg of SH per g of wet tissue.

### 2.6. Hg Determination

Hg levels were determined by inductively coupled plasma atomic emission spectrometry (ICPE-9000; Shimadzu Scientific Instruments, Kyoto, Japan). The samples of wet tissue were placed in vials and frozen at −20 °C until analysis. Samples were digested as previously described by Ineu et al. [49]. The analytical Hg standard (Merck^®^, Kenilworth, NJ, EUA) was used to make the curve. The detection limit was considered 0.0025 ppm.

### 2.7. Statistical Analysis

Results were analyzed by one-way ANOVA followed by Tukey’s multiple range test or Student’s *t*-test when appropriate. The effects were considered significant when *p* < 0.05. 

## 3. Results

### 3.1. Food, Water, and Hg^2+^ Intake

The total food and water intake and the Hg^2+^ intake estimation during the gestation and lactation periods are shown in Table 1. Hg exposure did not alter the total food intake during the gestation or lactation periods. On the other hand, one-way ANOVA showed a significant effect of treatment on total and daily water intake during the gestation [total: F(2,19) = 15.84; *p* ≤ 0.0001; daily: F(2,21) = 12.71; *p* = 0.0002] and lactation [total: F(2,19) = 4.619; *p* = 0.024; daily: F(2,21) = 5.309; *p* = 0.0136] periods. Dams exposed to 10 and 50 µg Hg^2+^/mL had a decrease in total water intake in both periods analyzed (gestation and lactation) when compared to control group (0 µg Hg^2+^/mL) (*p* < 0.05, Tukey’s multiple comparison test). The total Hg^2+^ intake was estimated based on total water intake. Student’s *t*-test showed that female rats exposed to 50 µg Hg^2+^/mL presented a higher total and daily Hg^2+^ intake during the gestation [total: t(15) = 11.84; *p* ≤ 0.0001; daily: t(15) = 12.54; *p* ≤ 0.0001] and lactation periods [total: t(15) = 8.081; *p* ≤ 0.0001; daily: t(15) = 13.56; *p* ≤ 0.0001] than animals exposed to 10 µg Hg^2+^/mL.

### 3.2. Body and Organ Weight

#### 3.2.1. Dams

Body weight and total and relative liver, kidney, and brain weight of dams exposed to 0, 10, and 50 µg Hg^2+^/mL during the gestation and lactation periods are shown in Table 2. One-way ANOVA showed a significant treatment effect on total [F(2,21) = 8.081; *p* = 0.0025] and relative [F(2,21) = 14.11; *p* = 0.0001] kidney weight. Dams exposed to inorganic Hg presented a dose-dependent increase in kidney weight (*p* < 0.05, Tukey’s multiple comparison test).

#### 3.2.2. Offspring

Pup body weight, as well as absolute and relative weights of liver, kidney, and brain on postnatal days 10, 20, 30, and 40 are shown in Table 3. Pups exposed to inorganic Hg during the pre- and postnatal periods did not present body weight alterations in the periods analyzed. Regarding organ weight, one-way ANOVA showed significant treatment effect on liver [F(2,18) = 5.33; *p* = 0.0152] and kidney [F(2,18) = 3.819; *p* = 0.0415] relative weight at PND 40. Pups exposed to 50 µg Hg^2+^/mL presented an increase in liver and kidney relative weight when compared the control group (0 µg Hg^2+^/mL) (*p* < 0.05, Tukey’s multiple comparison test).

### 3.3. Urea and Creatinine

#### 3.3.1. Dams

Serum urea and creatinine levels are shown in Table 4. Dams exposed to 10 and 50 µg Hg^2+^/mL in drinking water did not present alterations on serum urea and creatinine levels.

#### 3.3.2. Offspring

Like dams, pups exposed to inorganic Hg did not present alterations on serum urea and creatinine levels (Table 4). 

### 3.4. PBG-Synthase Activity 

#### 3.4.1. Dams

The liver, kidneys, and brain PBG-synthase activities are shown in Table 5. Mercury exposure in drinking water during the gestation and lactation periods did not alter the PBG-synthase activity of the different tissues evaluated.

#### 3.4.2. Offspring

As observed in dams, offspring exposed indirectly to inorganic Hg (*in utero* and breast milk) did not present significant PBG-synthase activity alterations in the different tissues and periods analyzed (Table 5). 

### 3.5. Thiol Groups

#### 3.5.1. Dams

Mercury exposure in drinking water during the gestation and lactation periods did not alter the total and non-protein thiol in the different tissues analyzed (data not shown). 

#### 3.5.2. Offspring

The offspring exposed to HgCl_2_
*in utero* and via breast milk did not present alterations in liver, kidney, and brain total and non-protein thiol (data not shown).

### 3.6. MT Levels

#### 3.6.1. Dams

Hepatic and renal MT levels are shown in Figure 1. Dams exposed to inorganic Hg (10 and 50 µg Hg^2+^/mL) presented an increase in hepatic [F(2,10) = 3.847; *p* = 0.0577] and renal [F(2,11) = 6.267; *p* = 0.0152] metallothionein levels when compared to the control group (0 µg Hg^2+^/mL).

#### 3.6.2. Offspring

Offspring exposed indirectly to HgCl_2_ (*in utero* and breast milk) did not present significant alterations in hepatic and renal MT levels (Table 6). 

### 3.7. Hg Levels

Figure 2 shows the renal Hg levels of dams exposed to HgCl_2_ during the gestation and lactation periods. One-way ANOVA showed effect of treatment in renal Hg content [F(2,9) = 201.4; *p* = 0.00001]. The exposure to HgCl_2_ caused a dose-dependent increase in renal Hg levels of dams (*p* < 0.05, Tukey’s multiple comparison test).

The Hg levels in the liver and brain of dams and liver, kidney, and brain of pups from different postnatal days were below the detection limit (data not shown). 

## 4. Discussion

In this work, we evaluated the toxicological effects of HgCl_2_-contaminated drinking water exposure during the gestational and lactational periods. Dams exposed to HgCl_2_ have a decrease in water intake during the gestation and lactation periods. As suggested in our previous work [41], water intake reduction might be caused by the Hg metallic taste [7]. Besides this, it is noteworthy to highlight that the Hg intake during the lactation period was about two times higher than during the gestational period. This result calls attention, as the inorganic Hg is the main Hg chemical form excreted through the milk [50,51].

Several studies have shown that HgCl_2_ exposure affects mainly the renal system [11,14,45,52,53]. The degree of renal damage is associated with the exposure model, time of exposure, and how long after the exposure the effects are evaluated. In our work, dams, and pups (PND 40) exposed to 50 µg Hg^2+^/mL have relative renal weight increased; however, the well-established markers of nephrotoxicity, urea and creatinine, were not altered. Several studies described the increase in renal weight followed by an increase in serum urea and creatinine levels in young [53,54], adult [11,55], and old [52,56] rats exposed subcutaneously or intravenously to HgCl_2_. Thus, we hypothesize that the amount of Hg burden on the kidney is causing proximal tubule damage. However, this damage was not enough to cause a deficit in the clearance of metabolic wastes, for instance, urea and creatinine. This may be explained because, during the gestational and lactational periods, the glomerular filtration rate and renal plasma flow are enhanced [57,58]. Still, the absence of biochemical alteration in dams and offspring exposed to HgCl_2_ could be explained by the affinity of Hg ions for the thiol groups of scavenger molecules, such as reduced glutathione and metallothionein, forming inert complexes [59]. Similarly, in our work, dams exposed to Hg presented a significant increase in renal metallothionein content concomitantly with an increase in renal Hg levels. Moreover, the hepatic metallothionein levels are slightly higher when compared with the control group in dams exposed to 10 and 50 µg Hg^2+^/mL; this is an expected result since the liver is one of the organs responsible by metallothionein synthesis [60].

Although Hg can bind to thiol groups of scavenger molecules forming an inert complex, it also can bind to extracellular and intracellular biomolecules that contain -SH groups. This can cause alterations in the conformational structure with consequent loss of function, leading to a cellular death following by a possible organ failure [18,61,62]. Enzymes are sensitive to Hg-thiol binding; for example, Hg is known to inhibit the enzymes lactate dehydrogenase [63] alanine aminotransferase [53,54], and PBG-synthase [45,64].

PBG-synthase is a sulfhydryl enzyme that participates in the heme synthesis pathway [44]. Usually, this enzyme is inhibited in different tissues after Hg exposure. In this study, dams and pups did not present PBG-synthase activity alteration in the liver, kidney, and brain. Probably, the amount of Hg that is reaching the tissues is not enough to inhibit the enzyme. The levels of Hg in the liver and brain from dams and liver, kidney and brain from pups were under the detection limit. Although Hg was detected in the kidney of dams, the PBG-synthase was not inhibited, maybe because the Hg ions were bound to the metallothionein molecules forming an inert complex. The absence of Hg detection in several organs was expected because the absorption of Hg^2+^ in the gastrointestinal system is not high [19] and few Hg ions are reaching the bloodstream. However, this method of exposure is very important to be studied. When using intravenous exposure, our research group was able to demonstrate the Hg accumulation in rats’ fetal tissues [42].

In conclusion, the exposure to HgCl_2_ in drinking water during the gestational and lactational periods caused alterations mainly to dams exposed to 50 µg Hg^2+^/mL. These alterations are related to the renal system, confirming the HgCl_2_ nephrotoxicity. Despite the Hg exposure has caused some renal damage; this damage was slight when compared with other studies. As inorganic Hg was administrated in drinking water for a long period, we believe that the dams’ body was able to handle the Hg ions, avoiding damage effects and protecting the offspring; it is probably related to the scavenger molecules, reduced glutathione and metallothionein. It is important to emphasize the importance of further studies to contribute to the explanation of some findings in this study, as well as to understand how inorganic mercury acts during pregnancy and lactation periods.

## Figures and Tables

**Figure 1 toxics-10-00664-f001:**
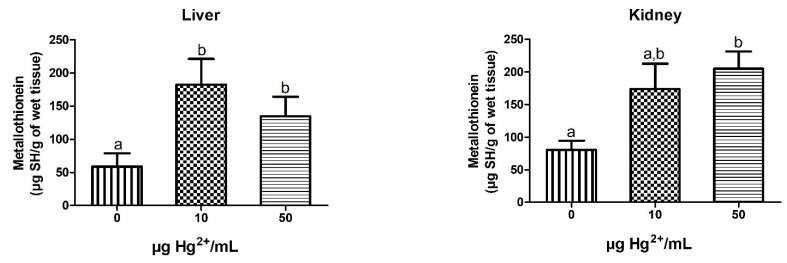
Hepatic and renal MT levels of dams exposed to different doses (0, 10, and 50 µg Hg^2+^/mL) of HgCl2. The results are presented as mean ± SEM. Groups with the same letters have no statistically significant difference and groups with different letters differ statistically (Tukey’s multiple range test; *p* < 0.05). 0 = 0 µg Hg^2+^/mL group (control group); 10 = 10 µg Hg^2+^/mL group; 50 = 50 µg Hg^2+^/mL group.

**Figure 2 toxics-10-00664-f002:**
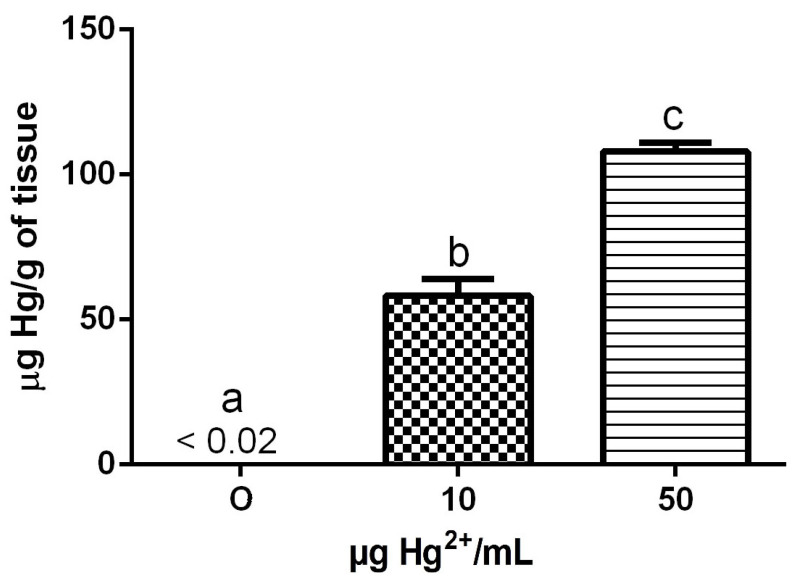
Renal Hg levels of dams exposed to different doses (0, 10 and 50 µg Hg^2+^/mL) of HgCl_2_ as described in Table 1. The results are presented as mean ± SEM. Groups with the same letters have no statistically significant difference and groups with different letters differ statistically (Tukey’s multiple range test; *p* < 0.05). 0 = 0 µg Hg^2+^/mL group (control group); 10 = 10 µg Hg^2+^/mL group; 50 = 50 µg Hg^2+^/mL group.

**Table 1 toxics-10-00664-t001:** Food, water, and Hg^2+^ intake during the gestation and lactation periods of dams exposed to different doses (0, 10, and 50 µg Hg^2+^/mL) of HgCl_2_ in drinking water.

	0 (*n* = 7)	10 (*n* = 9)	50 (*n* = 7)
**Food intake (g)**			
*Gestation*			
Total *	406.0 ± 48.1	438.4 ± 34.3	407.1 ± 43.3
100 g/day #	7.6 ± 0.7	8.1 ± 0.4	7.9 ± 0.8
*Lactation*			
Total *	945.5 ± 46.6	886.3 ± 41.3	906.3 ± 59.1
100 g/day #	16.5 ± 0.6	16.4 ± 0.6	17.6 ± 0.9
**Water intake (mL)**			
*Gestation*			
Total *	1001.0 ± 62.3 ^a^	717.1 ± 38.7 ^b^	608.3 ± 46.7 ^b^
100 g/day #	19.1 ± 1.4 ^a^	14.36 ± 0.5 ^b^	12.38 ± 0.9 ^b^
*Lactation*			
Total *	1752.0 ± 100.9 ^a^	1283.0 ± 95.3 ^b^	1247.0 ± 145.9 ^b^
100 g/day #	30.8 ± 1.9 ^a^	24.4 ± 0.9 ^b^	24.9 ± 1.8 ^b^
**Hg2+ intake (mg)**			
*Gestation*			
Total *	n.d.	7.03 ± 0.37	30.41 ± 2.33 ^$^
100 g/day #	n.d.	0.14 ± 0.00	0.62 ± 0.05 ^$^
*Lactation*			
Total *	n.d.	12.99 ± 0.87	62.33 ± 7.29 ^$^
100 g/day #	n.d.	0.24 ± 0.01	1.25 ± 0.09 ^$^

The results are presented as mean ± SEM. Groups with the same letters have no statistically significant difference and groups with different letters differ statistically (Tukey’s multiple range test; *p* < 0.05). ^$^ differs from group 10 (Student’s *t*-test; *p* < 0.05). n.d. = not determinate. 0 = 0 µg Hg^2+^/mL group (control group). 10 = 10 µg Hg^2+^/mL group. 50 = 50 µg Hg^2+^/mL group. * Total intake during the gestation or lactation period. # Food, Water or Hg^2+^ Intake Per 100 g of Body Weight per Day.

**Table 2 toxics-10-00664-t002:** Body weight, total and relative weight organs weight of dams exposed to different doses (0, 10, and 50 µg Hg^2+^/mL) of HgCl_2_ in drinking water during pregnancy and lactation period.

	0 (*n* = 7)	10 (*n* = 9)	50 (*n* = 7)
**Body weight** (g)			
Gestation day 0	221.30 ± 23.39	229.20 ± 12.78	225.50 ± 1457
Last day of gestation	318.30 ± 25.82	329.50 ± 19.12	308.50 ± 19.12
Last day of lactation	259.80 ± 21.80	247.30 ± 10.87	247.00 ± 10.68
**Liver**			
Total (g)	12.91 ± 0.79	12.34 ± 0.63	12.97 ± 0.78
Relative (%)	4.60 ± 0.22	4.60 ± 0.18	4.72 ± 0.14
**Kidney**			
Total (g)	2.01 ± 0.09 ^a^	2.20 ± 0.08 ^b^	2.57 ± 0.12 ^c^
Relative (%)	0.72 ± 0.03 ^a^	0.82 ± 0.03 ^b^	0.94 ± 0.03 ^c^
**Brain**			
Total (g)	1.67 ± 0.07	1.69 ± 0.04	1.66 ± 0.02
Relative (%)	0.60 ± 0.03	0.63 ± 0.02	0.62 ± 0.03

The results are presented as mean ± SEM Groups with the same letters have no statistically significant difference and groups with different letters differ statistically (Tukey’s multiple range test; *p* < 0.05). 0 = control group. 10 = 10 µg Hg^2+^/mL group. 50 = 50 µg Hg^2+^/mL group.

**Table 3 toxics-10-00664-t003:** Organs total and relative weight of pups exposed to inorganic Hg *in utero* and via milk from dams exposed to different doses (0, 10, and 50 µg Hg^2+^/mL) of HgCl_2_ in drinking water during the pregnancy and lactation period.

			Liver		Kidney		Brain	
		Body Weight (g)	Total (g)	Relative (%)	Total (g)	Relative (%)	Total (g)	Relative (%)
	0 (*n* = 7)	19.22 ± 1.33	0.55 ± 0.03	2.87 ± 0.07	0.24 ± 0.01	1.28 ± 0.03	0.87 ± 0.04	4.59 ± 0.25
PND10	10 (*n* = 9)	17.11 ± 1.16	0.47 ± 0.02	2.81 ± 0.19	0.22 ± 0.01	1.31 ± 0.05	0.86 ± 0.03	5.16 ± 0.24
	50 (*n* = 7)	18.62 ± 0.78	0.51 ± 0.03	2.73 ± 0.09	0.25 ± 0.02	1.34 ± 0.04	0.85 ± 0.02	4.60 ± 0.20
	0 (*n* = 7)	41.21 ± 2.11	1.37 ± 0.07	3.34 ± 0.12	0.63 ± 0.12	1.50 ± 0.24	1.17 ± 0.09	2.89 ± 0.28
PND20	10 (*n* = 9)	35.92 ± 2.03	1.24 ± 0.10	3.43 ± 0.16	0.53 ± 0.11	1.43 ± 0.20	1.18 ± 0.06	3.37 ± 0.26
	50 (*n* = 7)	39.86 ± 2.35	1.37 ± 0.10	3.41 ± 0.11	0.74 ± 0.16	1.76 ± 0.29	1.18 ± 0.08	3.07 ± 0.32
	0 (*n* = 7)	77.11 ± 4.22	3.39 ± 0.16	4.42 ± 0.13	0.87 ± 0.05	1.12 ± 0.02	1.46 ± 0.02	1.93 ± 0.10
PND30	10 (*n* = 9)	61.53 ± 7.68	2.84 ± 0.34	4.67 ± 0.19	0.77 ± 0.06	1.33 ± 0.11	1.35 ± 0.05	2.40 ± 0.31
	50 (*n* = 7)	68.70 ± 5.50	3.10 ± 0.20	4.64 ± 0.32	0.90 ± 0.05	1.37 ± 0.11	1.40 ± 0.03	2.16 ± 0.23
	0 (*n* = 7)	138.10 ± 7.23	5.93 ± 0.35	4.29 ± 0.08 ^a^	1.52 ± 0.05	0.93 ± 0.03 ^a^	1.52 ± 0.05	1.11 ± 0.05
PND40	10 (*n* = 9)	117.20 ± 7.05	5.29 ± 0.49	4.48 ± 0.16 ^a,b^	1.49 ± 0.06	0.98 ± 0.01 ^a,b^	1.49 ± 0.06	1.28 ± 0.05
	50 (*n* = 7)	133.10 ± 4.96	6.40 ± 0.24	4.82 ± 0.12 ^b^	1.50 ± 0.03	1.01 ± 0.02 ^b^	1.50 ± 0.03	1.13 ± 0.04

The results are presented as mean ± SEM. Groups with the same letters have no statistically significant difference and groups with different letters differ statistically (Tukey’s multiple range test; *p* < 0.05). PND: Postnatal day. 0 = control group. 10 = 10 µg Hg^2+^/mL group. 50 = 50 µg Hg^2+^/mL group.

**Table 4 toxics-10-00664-t004:** Serum urea and creatinine levels from dams and pups exposed to different doses (0, 10, and 50 µg Hg^2+^/mL) of HgCl_2_ as described in Table 1 and Table 3, respectively.

		Urea (mg/dL)	Creatinine (mg/dL)
Dams			
	0 (*n* = 7)	74.30 ± 2.33	0.47 ± 0.09
	10 (*n* = 7)	77.01 ± 3.21	0.52 ± 0.11
	50 (*n* = 7)	64.76 ± 3.27	0.53 ± 0.07
PUPS			
	0 (*n* = 7)	44.00 ± 2.30	0.46 ± 0.09
PND10	10 (*n* = 7)	46.15 ± 5.99	0.30 ± 0.08
	50 (*n* = 7)	47.99 ± 4.34	0.30 ± 0.04
	0 (*n* = 7)	42.99 ± 2.46	0.72 ± 0.26
PND20	10 (*n* = 7)	55.90 ± 7.96	0.74 ± 0.27
	50 (*n* = 7)	57.41 ± 5.18	0.72 ± 0.25
	0 (*n* = 7)	49.24 ± 5.74	0.38 ± 0.03
PND30	10 (*n* = 7)	56.31 ± 6.64	0.34 ± 0.07
	50 (*n* = 7)	46.46 ± 4.85	0.39 ± 0.05
	0 (*n* = 7)	47.27 ± 4.26	0.19 ± 0.03
PND40	10 (*n* = 7)	57.13 ± 7.02	0.17 ± 0.04
	50 (*n* = 7)	38.66 ± 4.53	0.34 ± 0.06

The results are presented as mean ± SEM. PND: Postnatal day. 0 = control group. 10 = 10 µg Hg^2+^/mL group. 50 = 50 µg Hg^2+^/mL Group.

**Table 5 toxics-10-00664-t005:** Porphobilinogen synthase activity from different organs of dams and pups exposed to different doses (0, 10, and 50 µg Hg^2+^/mL) of HgCl_2_ as described in Table 1 and Table 3, respectively.

		Liver	Kidney	Brain
**Dams**				
	0 (*n* = 7)	11.16 ± 1.76	5.81 ± 0.93	1.72 ± 0.33
	10 (*n* = 7)	10.74 ± 0.96	6.19 ± 1.32	1.41 ± 0.30
	50 (*n* = 7)	17.21 ± 2.90	8.51 ± 1.24	1.87 ± 0.30
**Pups**				
	0 (*n* = 7)	21.39 ± 7.97	9.16 ± 1.93	1.23 ± 0.34
PND10	10 (*n* = 7)	20.12 ± 4.36	7.65 ± 1.08	1.76 ± 0.46
	50 (*n* = 7)	22.87 ± 6.49	9.43 ± 2.05	1.36 ± 0.34
	0 (*n* = 7)	19.78 ± 2.77	7.04 ± 0.87	1.64 ± 0.29
PND20	10 (*n* = 7)	22.40 ± 2.80	6.59 ± 1.00	1.35 ± 0.21
	50 (*n* = 7)	24.18 ± 3.56	10.37 ± 1.48	1.67 ± 0.25
	0 (*n* = 7)	21.75 ± 3.95	9.77 ± 0.90	1.13 ± 0.23
PND30	10 (*n* = 7)	17.73 ± 2.45	9.42 ± 1.87	0.95 ± 0.24
	50 (*n* = 7)	21.54 ± 2.67	9.73 ± 1.34	1.20 ± 0.21
	0 (*n* = 7)	16.16 ± 1.77	8.03 ± 0.46	1.15 ± 0.15
PND40	10 (*n* = 7)	23.20 ± 2.39	8.64 ± 0.28	1.13 ± 0.11
	50 (*n* = 7)	16.84 ± 1.45	8.15 ± 0.59	1.04 ± 0.10

The results are presented as mean ± SEM. The enzymatic activity is expressed as nmol PBG/h/mg protein. PND: Postnatal day. 0 = control group. 10 = 10 µg Hg^2+^/mL group. 50 = 50 µg Hg^2+^/mL Group.

**Table 6 toxics-10-00664-t006:** Metallothionein from different organs of pups’ rats exposed to different doses (0, 10, and 50 µg Hg^2+^/mL) of HgCl_2_ as described in Table 1 and Table 3, respectively.

		Liver	Kidney
PUPS			
	0 (*n* = 5)	96.76 ± 24.33	117.60 ± 21.31
PND 10	10 (*n* = 4)	147.80 ± 37.15	217.00 ± 58.50
	50 (*n* = 5)	95.20 ± 16.92	113.10 ± 31.08
	0 (*n* = 5)	90.49 ± 24.70	84.02 ± 33.51
PND 20	10 (*n* = 4)	110.30 ± 29.88	122.00 ± 34.83
	50 (*n* = 5)	120.20 ± 27.05	138.10 ± 22.69
	0 (*n* = 5)	62.65 ± 18.91	66.24 ± 18.91
PND 30	10 (*n* = 4)	88.86 ± 20.55	95.44 ± 33.31
	50 (*n* = 5)	56.16 ± 19.30	61.83 ± 23.57
	0 (*n* = 5)	89.55 ± 24.70	85.72 ± 27.94
PND 40	10 (*n* = 4)	81.36 ± 26.62	92.07 ± 21.36
	50 (*n* = 5)	58.40 ± 20.45	63.39 ± 18.70

The results are presented as mean ± SEM. Groups with the same letters have no statistically significant difference and groups with different letters differ statistically (Tukey’s multiple range test; *p* < 0.05). The metallothionein levels are expressed as µg SH/ g of wet tissue. PND: Postnatal day. C = control group. 10 = 10 µg Hg^2+^/mL group. 50 = 50 µg Hg^2+^/mL group.

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
