# Peer review of "Biochemical Parameters of Female Wistar Rats and Their Offspring Exposed to Inorganic Mercury in Drinking Water during the Gestational and Lactational Periods"

_toxics, 2022, doi:10.3390/toxics10110664_

Round 1

Author Response

Kind regards. 

Reviewer 2 Report

This is an interesting article. However, my main concern is the poor discussion of the main findings. I consider that authors should explain better (and justify) their findings. The English written is correct, but some sentences would need to be reviewed for easy understanding. Please consider my comments below:

ABSTRACT

Authors should consider adding some values to their main findings so they are properly justified.

MATERIALS AND METHODS

Authors should justify why they used Wistar rat for their study.

Ten rats, is this number enough to support their study? This should be explained as authors indicate that some rats could not get pregnant/have offspring during their experiments, so different numbers of subjects are presented for each group (e.g., 10 and 7).

Authors should briefly explain the selection of Hg concentrations/doses and not only referring to a previous manuscript.

RESULTS AND DISCUSSION

Authors describe that the levels of Hg in pups were all below the limit of detection. This should be explained in the discussion section and compared with previous studies reported in the literature. Is this a common finding?

Moreover, authors provide different numbers of specimens in the different tables, which suggest that some subjects are missing or died? Could this be explained/indicated?

The overall discussion section needs to be comprehensive reviewed. Authors should explain all their findings and propose potential hypothesis that could explain their findings. First paragraph should be removed as it is not discussion. Important findings such as the increase in Hg intake during lactation and why this did not result in an increase in the content of Hg in pups should be explained.

The presence of Hg below the limit of detection in different organs in dams and pups should be also discussed and explained in more detail.

Moreover, all findings should be compared with other similar studies reported in the literature. Gaps of knowledge should be indicated. Authors should suggest some potential lines of research to fill in these gaps.

Author Response

Kind regards. 

Round 2

Reviewer 2 Report

Dear authors, thank you for addressing my comments. I think that the revised version is more robust and scientifically sound. There are a few things that would benefit your manuscript, specially about the levels of Hg detected in pups (all below the limit of detection). This should be explained and compared with previous studies reported in the literature, so the reader can know if this is a common finding or something to investigate further.  

Author Response

Response to Reviewer 2 Comments

Dear authors, thank you for addressing my comments. I think that the revised version is more robust and scientifically sound. There are a few things that would benefit your manuscript, specially about the levels of Hg detected in pups (all below the limit of detection). This should be explained and compared with previous studies reported in the literature, so the reader can know if this is a common finding or something to investigate further.

Response:      Thank you. We improve the discussion about the Hg levels in pups, pointing out that when another exposure via was used, we were able to detect Hg in rats' fetal tissues. We hope now our paper is suitable for publication.

“The absence of Hg detection in several organs was expected because the absorption of Hg2+ in the gastrointestinal system is not high (Lorschieder et al., 1995) and few Hg ions are reaching the bloodstream. However, this via of exposure is very important to be studied. However, when using intravenous exposure, our research group was able to demonstrate the Hg accumulation in rats' fetal tissues (Oliveira et al. 2015).